# Offshore seismicity clusters in the West Iberian Margin illustrated by two decades of events.

Gabriela Fernandez-Viejo[1], Carlos Lopez-Fernandez[1], Patricia Cadenas[2]

[1]Department of Geology, University of Oviedo, Jesús Arias de Velasco s/n 33005 Oviedo, Spain
[2]Memorial University of Newfoundland, Dept. Earth Sciences

*Correspondence to*: Gabriela Fernandez-Viejo (fernandezgabriela@uniovi.es)

**Abstract.** An analysis of two decades (2003-2022) of seismicity recorded by the Spanish and Portuguese seismic networks along the West Iberian passive margin has resulted in a better understanding of the distribution of moderate seismic activity in this intraplate submarine area. The study provides a precise trend of specific alignments inferred from the density maps of seismicity, giving an accurate depiction of event distribution along two wide stripes that extend 700 km long through the ocean floor in a WNW-ESE direction. These bands are parallel to the Africa-Eurasia plate boundary but are distinctly separated from its related seismicity by approximately 300 km and 700 km, respectively. This is a sufficient distance to be considered as intraplate activity. When trying to relate this seismicity to structural and geophysical features, a conclusive picture doesn't emerge. The earthquakes occur indiscriminately across thinned continental, hyperextended, and exhumed mantle rift domains. They fade out in the proximity of undisputed oceanic crust, but some events extend beyond. The hypocentral depths signal a considerable amount of events nucleating in the upper mantle. The focal mechanisms, although scarce, are predominantly strike-slip. Considering these observations, hypothesis ranging from subduction initiation, development of strained corridors or local structures of the margin, have been discussed in order to explain this relatively anomalous seismicity. However, some of them do not portray convincing arguments, while others are too unspecific. None of them are flawless, suggesting that several factors may be at play. Despite being one of the most probed passive margins in the world, the present geodynamical status of the West Iberian Margin manifested in its modern seismicity remains unknown. Interpreting this data within a global tectonic plate framework, together with the potential addition of seafloor seismometers, may provide the key to understanding this activity along one of the most archetypical margins of the Atlantic Ocean.

## 1 Introduction

### 1.1 Physiographic aspects and brief geological history of the West Iberian Margin

The West Iberian Margin (WIM) is an 850 km long passive margin parallel to the N-S trending rectilinear coast of western Iberia that resulted from a tectonic evolution marked by a multiphase rifting history (Pinheiro et al., 1996; Péron-Pinvidic et al., 2007; Tucholke et al., 2007; Alves et al., 2009; Pereira et al., 2016; Granado et al., 2021). Currently, the dynamics of the margin are influenced by two significant plate limits. To the west lies the mid-Atlantic ridge (MAR) constructive plate

boundary. Meanwhile, to the south, the margin is impacted by the complex NW Africa-SW Eurasia plate limit, known as the Azores-Gibraltar Fault Zone (AGFZ) (Fig. 1). To the north, the West Iberian Margin (WIM) adjoins the E-W trending southern Biscay passive margin, which locates to the south of the extinct Bay of Biscay ridge.

Along the WIM, the 30-85 km wide continental platform is noticeably intersected by deep E-W/NE-SW trending canyons, namely Nazaré (NZc), Setubal (STc), and San Vicente (SVc) (Fig. 1). Northwards of NZc, 200 km from the coast and separated from the platform by the Galicia Interior Sedimentary Basin (GIB), the Galicia Bank (GB) stands out, which is an elevated area with a relief of over 3,500 m, along with other minor topographic seamounts. Westwards of the GB, the Iberian Abyssal Plain (IAP) emerges as an expansive and flat ocean floor that reaches a depth of 5,300 m. As we move towards the MAR, the surface becomes considerably rough, indicative of oceanic crust. To the west of the IAP, the prominent Kings Through (KT) is a 450 km long and WNW-ESE trending scar in the oceanic floor (Fig. 1) (Srivastava and Roest, 1992), which consists of a central depression flanked by ridges. It is proposed that this feature formed from 56 to 25 Myr during the Mesozoic rifting along the crest of a hotspot, resulting in an aseismic ridge, which represented the plate boundary between Iberia and Eurasia. This boundary shifted and was active along the Bay of Biscay between 49-36 Myr, extending eastwards along the Pyrenees (Roest and Srivastava, 1991) and finally moved to its present location along the AGFZ at Chron 13 (Macchiavelli et al., 2017). Between NZc and STc, the Extremadura Spur (Es) separates the IAP from the Tagus Abyssal Plain (TAP), a 200 km wide flat area and situated at 5,000 m depth. The TAP is limited to the west by the Madeira Tore Rise and to the southeast by the Gorringe Bank, one of the principal seismogenic sources in the region. The presence of these reliefs has been associated with N-S normal faults and W-E thrusts (Boillot et al., 1988a, Pinheiro et al., 1996; Alvarez-Marron, et al., 1997, Vazquez et al., 2008; Duarte et al., 2013; Sallarès et al., 2013), which in turn can be related to the margin reactivation during the Mesozoic and the Paleogene (e.g., Jiménez-Munt y Negredo, 2013). Furthermore, the Extremadura Spur and related reliefs are connected to extensive Late Cretaceous alkaline magmatic occurrences (Pereira et al., 2016; Escada et al., 2022).

The mid-Atlantic ridge (MAR), a constructive plate boundary situated about 1,500 km west of the WIM, separates the North American plate from the Eurasian plate. With a slow-spreading rate, it accretes at a rate of 24-26 mm/y at 43ºN (Somoza et al., 2019). Unlike other areas of the North Atlantic, the MAR is remarkably devoid of significant transform faults, spanning over 1400 km without any such faults between 40.6ºN and 43.6ºN and between 43.6ºN and 47.6ºN, except for short ones with negligible displacement. This unique characteristic of the ridge in this section of the North Atlantic contrasts with its displacement by relevant transform faults both to the south and north of the WIM.

The AGFZ plate boundary that limits the WIM southwards is structurally much more complex. The Southwest Iberian Margin (SWIM) faults (Zitellini et al., 2009; Martínez-Loriente et al., 2013) connect the Gloria Fault with the eastern part of the Africa-Europe plate boundary (Fig. 1). The AGFZ is sometimes described as a diffuse convergent plate boundary to the East of Gibraltar due to the ample area of deformation. But a recent seismic study interprets long and active faults accommodating most of the deformation in the Western Mediterranean (Gomez de la Peña et al., 2022)

The kinematic evolution along theWIM abyssal plain is well determined by magnetic anomalies after the Cretaceous normal polarity superchron (Roest and Srivastava, 1991) However, the early rift to drift seafloor evolution from 200 Myr to 83 Myr,

remains controversial, mainly due to the dual assignment of the J anomaly either to a real isochron (e.g., Barnett-Moore et al., 1996) or to a polygenic magmatic event right before oceanic inception (e.g., Nirrengarten et al., 2017).

### 1.2 Crustal structure and domains of the WIM

The WIM fits into the category of a non-volcanic passive margin with a narrow platform (Lymer and Reston, 2022). It developed conjugate to the Newfoundland margin during the Mesozoic rifting (Péron-Pinvidic and Manatschal, 2009). Its crustal structure, although debated in minor aspects, is formed by a series of structural domains from East to West (e.g., Péron-Pinvidic and Manatschal, 2009; Ramos et al., 2016; Druet et al., 2019; Merino et al., 2021) and includes a Variscan basement (Schulmann et al., 2022). The segmentations observed within the WIM resulted from the northward migration of extension and successive late Jurassic to early Cretaceous extensional phases (e.g., Malod and Mauffret, 1990; Péron-Pinvidic et al., 2007; Tucholke et al., 2007; Merino et al., 2021).

Following the onset of breakup within the Pangea supercontinent 200 Myr ago, the pre-Jurassic template of the North Atlantic was subjected to three main phases of rifting that took place from Late Triassic to Early Jurassic, Late Jurassic to Early Cretaceous, and during Mid Cretaceous. The different rifting stages led to a system of tilted blocks limited by westward-dipping extensional detachment faults (Lymer and Reston, 2022), which show a N-S orientation in the North and variable trends in the South. At the end of rift-related deformation, during the Late Cretaceous, the WIM experienced continental break-up (Soares et al., 2012; Welford and King, 2021), leading to irregular regions of exhumed and serpentinized mantle (Afilhado et al., 2008; Merino et al., 2021; Grevemeyer et al., 2022). The onset of seafloor spreading occurred at least in two pulses, a first one at the end of the Jurassic between the Grand Banks and the TAP, and a second one during the early Cretaceous between the IAP and the Grand Banks, following the temporal northward opening of the Atlantic Ocean.

As a consequence of the later Alpine orogeny that formed the actual reliefs on land, there was an important compressive deformation, which produced the reactivation of some extensional faults as thrusts, or as transpressive strike slip faults (e.g., Pereira et al., 2011; Martínez-Loriente et al., 2013; Ramos et al., 2016; Terrinha et al., 2019). The Cenozoic deformation abruptly terminates against the Ocean-Continent transition (OCT).

Published works investigating the Mesozoic rift domains in the WIM distinguish two sectors. In the northern sector, the continental hyperextended crust stretches for more than 300 km and the exhumed mantle domain extends westwards of the GB (Druet et al., 2019). In the southern sector, contrastingly, the hyperextended crust extends for 200 km before reaching the same transition (Pedrera et al., 2017). The OCT extends between 12º10’W and 12º30’W in the IAP (Whitsmarsh et al., 1990) and it would extend Nº10 for 130 km until Extremadura Spur. According to Grevemeyer et al. (2022), the OCT would run between 12.5ºW and 13ºW and in the TAP, while Merino et al. (2021) locate the OCT between 13-13.5ºW.

Based on seismic-derived gravity modelling, Amigo Marx et al. (2022), differentiate a 18 km thick proximal domain, a <18 km thick extended domain, and a < 7km thick hyperextended domain. The exhumed mantle domain has been drilled at a serpentinite ridge (Boillot et al., 1988b). Westwards of the Extremadura Spur, a thin lower crust has been interpreted beneath a dense upper crust. The thickness of the crust decreases quickly towards the West, while it does so gradually towards the

North, creating two distinct domains. The first one is located near the continent and characterized by a thin and light crust, while the second one is further away and has a denser and thicker crust.

Using seismic, wells and gravity data, Granado et al. (2021) modelled a more complex 3D crustal scale structure of the WIM. with two necking domains along the GIB and the DGM, suggesting two rift directions. The GIB encompasses localized regions of significantly thinned crust that are segmented by NE-SW trending transfer zones and it has been interpreted as a failed rift (Murillas et al., 1990; King et al., 2020). The GB is 15-20 km thick (Péron-Pinvidic and Manatschal, 2010; King et al., 2020). The North has a narrower exhumed mantle domain than the southern WIM. The outer necking zone merges southwards with the eastern one following a NW-SE orientation. The southern sector has a narrow continental domain, with a broader exhumed mantle domain.

The limit between the North and South WIM, follows the NW-SE orientation observed in the Bouguer gradient anomaly maps THD and dZ. According to Granado et al. (2021) it relates to the Variscan fabric onshore, suggesting unrecognized transfer structures to explain the transition. A similar transfer zone SW of the GB was suggested earlier (Pinheiro et al., 1992; Whitsmarsh et al., 1993) based on the analysis of magnetic anomalies and named as the Figueira Fracture zone, which would represent an original fracture zone born at the time of initiation of seafloor spreading.

So, within the general dichotomy North WIM-South WIM, we can consider in our study area two of the three WIM segments from north to south: the Galicia margin that includes the GB and IAP, and the southern segment that includes the Extremadura Spur and the TAP. The seismicity related to the third and southernmost segment of the WIM, the so-called Southwest Iberian Margin is out of the scope of this study, as it is considered related to the AGFZ plate boundary (Buforn et al., 1988).

### 1.2 Stress state, seismicity and kinematics in the WIM

Seismicity is the expression of active stresses within plates, and these forces are intimately linked to kinematics. The so-called Iberian plate takes its name from periods where effectively, the Iberian Peninsula and part of its margins acted as an independent minor tectonic plate between the surrounding and larger European and African plates (e.g., Roest and Srivastava, 1991). At earlier periods, Iberia was attached to Africa, having a plate boundary running in the northern part through the Bay of Biscay and the actual Pyrenees, and subsequently, moved in solidarity with Europe. The presence of thick post-orogenic and post-tectonic units offshore sealing the Biscay accretionary wedge (Álvarez-Marron et al., 1997; Fernández-Viejo et al., 2012) among other things confirm that Iberia is now part of the Eurasian plate without discussion. Nevertheless, Iberian kinematics have been for a long time a cause of study and controversy. The M sequence of magnetic anomalies along the Newfoundland and WIM forms one of the primary constraints for plate kinematic reconstructions characterising the motion of Iberia during the Mesozoic (Vissers and Mejer, 2012; Sibuet et al., 2004, Barnett-Moore et al., 2016 and references therein). However, the validity of these anomalies has been questioned (Bronner et al., 2011). The difficulty in distinguishing different crustal types only adds uncertainty (Granado et al., 2021, and references therein). The seafloor spreading history of the western and northern margins of Iberia has been described involving successive jumps of the plate boundary between Iberia, Europe, and Africa from north (Bay of Biscay axis, and KT) to south (AGFZ) (Roest and Srivastava, 1991; Macciavelli et al., 2017).

Complexity in kinematics is normally inversely proportional to the size of the plate. Therefore, numerous models of Iberia whereabouts have been published in the last decades, the most modern ones approaching the problem through deformable plate tectonic models (King and Welford, 2022). These have provided insight into the kinematic role of continental blocks and their interplay with large and micro-tectonic plates during the formation of the north Atlantic rifted margins (e.g., Nirrengarten et al., 2018; Peace et al., 2019; King and Welford, 2022).

Macchiavelli et al. (2017) indicates a spreading rate of 23 mm/yr north of 40ºW latitude and 20.5 mm year south of 38 ºW and 2 mm/yr of N-S convergence between Iberia and Africa since the Miocene. The earlier partitioning of convergence between north and south became totally accommodated by the southern boundary from early Miocene onwards. The total convergence oscillates between 3-6 mm/y (Fernandes et al., 2003; Nocquet, 2012; Serpelloni, et al., 2004). It is characterized around the Azores archipelago by an NNE-SW extension that passes to the East to a dextral strike-slip border along the Gloria fault. Along the Gulf of Cádiz becomes an obliquely convergent boundary, N45ºW (Ribeiro et al., 1996; Stich et al., 2006; Pedrera et al., 2011) with a velocity of 4.5 mm/year (Nocquet and Calais, 2004; Stitch et al., 2006), that accommodates intense deformation along shear zones (Terrinha et al., 2009), and through the reactivation of thrusts in the SWIM (Ramos et al., 2016). A neotectonic model across the southern plate boundary indicates that friction coefficients between 0.06 and 0.1 are the best solutions and that the maximum slip rates predicted outside the ridge occur along the Gloria fault, which is almost devoid of seismicity (Jimenez-Munt and Negredo, 2003).

Geodetic velocities (Garate et al., 2015, Palano et al., 2015) also reveal an important deformation in the southern sector of Iberia, while in the interior of the Iberian Peninsula the crustal deformation is locally accommodated, with rates of less than 15 nanostrain/yr. The stations situated in central and northern Portugal are displaced to the north around 1 mm/yr (Palano et al., 2015). Along the NW margin, the geodetical data evidence E-W contraction, with rates of 55 nanostrain/yr. In the WIM, there is movements to the NW reaching 3 mm/year. In the active stress map of the Iberia, the transition from a compressive to a extensive regime occurs just around the Lisboa-Nazaré latitude (Fig. 1). A peculiarity worth to note is the large amount of compression that the map shows in this area and the radical change to extension north of this transition.

Within the global picture, the present-day NNW-SSE compressional stress, has been practically constant since the Miocene (Andeweg et al., 1999; de Vicente et al., 2008 and references therein), and oriented at high angles to both north and west coasts of Iberia, being subparallel to the tensile stresses induced by the lateral density variations along these Iberian margins (Andeweg et al., 1999).

These complex system of forces and movements outlined above is the engine producing the intraplate earthquakes in the WIM. Seismicity in the southernmost WIM is overwhelmingly clustered, as expected on both plate boundarues, along the AGFZ, with significantly complex patterns, and along the MAR. However, within mainland Western Iberia and along adjacent offshore domains, away from both boundaries, seismicity has been repeatedly classified as diffuse. There is a general consensus on noting that the magnitude, and the number of events are difficult to reconcile with a typical intraplate location (Custodio et al., 2015; Kruger et al., 2020; Veludo et al., 2017; Duarte et al., 2013; Ribeiro, 2002).

The seismicity at the WIM hasn't been directly approached in published studies, and the absence of Ocean Bottom Seismometers except for some local studies (Kruger et al., 2020) has discouraged a profound study on their characteristics. Custodio et al. (2015) indicated that the marine earthquakes do collapse into well-defined clusters while cumulative seismic moment and epicentre density decrease from South to North. The analysis of some focal mechanisms corroborates that Portugal is under horizontal pressure in the NNW-SSE direction, with a great proportion of strike-slip solution and some reverse oblique mechanisms onshore (Borges et al., 2001).

## 2 Gathering seismicity in the WIM

The data gathered for this study belong to the permanent seismic networks of Portugal and Spain. In the Portuguese network, operated by the Instituto Português do Mar e da Atmosfera (IPMA) there is a network of 50 land stations, 5 in Madeira archipelago and another 35 in Azores. The Spanish network, operated by the Instituto Geográfico Nacional de España (IGN), maintains 60 stations on land and 56 in the Canary Islands. We have retrieved data from the available catalogues since January 2003 until December 2022 in the area between 38ºN and 45ºN and 8ºW and 20º W, showing magnitudes > 2.5.

In total, 278 events were collected from the IPMA network and 649 from the IGN network. Integrating both catalogues, and selecting, in case of coincidence, those with the lesser error in location, we elaborated a final working catalogue of 708 events. From these, we stablished several requisites of quality required to be included in this study (RMS < 1, Smax < 25, Smin < 25 and Err < 75), resulting in 606 events, 352 from the Spanish network and 254 from the Portuguese network. (Table 1).

Likewise, we have included the existing historical seismicity data in the IGN catalogues and collected in the work of Ferrao et al. (2016), which contains information on 10 historical events that occurred within the study area. The 9 focal mechanisms considered in this work have been obtained from the CMT Catalogue (Dziewonski et al., 1981; Ekström et al., 2012) and from the IGN Seismic Moment Tensor Database (Rueda and Mezcua, 2005).

The information collected was integrated through a geographic information system with (i) bathymetric information provided by the European Marine Observation and Data Network (EMODnet Bathymetry Consortium, 2020), (ii) geological structures mapped in the continental platform (Somoza et al., 2021), (iii) gravity and magnetic data (Granado et al., 2021, GeoMapAPP-Ryan et al., 2009), and (iv) data from crustal Moho Depth (Granado et al., 2021; Whitmarsh, 1990; GeoMapApp - Ryan et al., 2009). The ArcMap v.10.3.1 (ESRI) program was used.

## 3 Results, clusters and trends

### 3.1 Distribution, magnitudes and focal mechanisms of the events in the WIM

Figure 2 displays the concentration of activity over the last 20 years, which extends from the coast to 18ºW. The figure shows the distribution of events collected and processed, represented on a marine bathymetric chart (Fig. 2a), along with a density map of events (Fig. 2b) that has been calculated.

According to those results, events are more abundant at the coast, blending in with the onshore, more dispersed seismicity. Going seawards, we start to recognise a subtle clustering, which seems to define two separated diffuse bands that extend up to 18ºW longitude and become narrower with distance. Further up towards the MAR, there is some isolated events. The orientation of these bands is about N80ºW. Although these blurred alignments have been inferred before (Custodio et al., 2015) the discrimination of events made in this study permits to get rid of badly located earthquakes that add to fuzziness giving place to a sharper image of these two seismic alignments. However, the separation in two bands is not that simple.

Figure 3 represents the depth distribution of events, superimposed to crustal domain cross-sections representative of the latitudes of the alignments (Granado et al., 2021). The bands observed are named in this study as the Galicia (N) and the Lisbon one(S). Specifically, they can be described by particular characteristics:

a). **Events offshore Galicia, WIM (North)**. This density band of events is oriented N75ºW from the coast until 17.5ºW. It spreads over 670 km and ends in the proximity of the bottom topographic expression of the Azores-Biscay rise (Fig. 2a). It seems there is a clustered seismicity in two main spots along the band with a middle gap.

Offshore in this sector during the studied period, we registered a total of 165 events over 2.5, (8 events/yr), and 21 over 4, (1/yr). Most of them are over 20 km depth, and progressively shallower to the west. The depths of events include mantle depths below the geophysical inferred Moho depth, which rises from 22 km in the coast up to 11 km oceanwards, at the end of the lineation. The β-value is 0.87.

In profile 1 (Fig. 3), which represents the crustal cross-section type for the Galicia Margin, there is an abundance of events in the thickest crustal areas, corresponding to the GB or to the proximal domain. The number of events is larger at the transition between hyperextended crust and exhumed mantle. There is an arguably but noticeable 50 km wide gap in event distribution west of the Galicia Bank, followed by a second cluster in the distal margin, which coincides with an area of topographic roughness in the ocean bottom.

The figures also show that there is a non-negligible percentage of events inside the uppermost mantle, especially within the transition between the hyperextended and exhumed mantle domains. A particular set of south-dipping earthquakes can be observed in profile 5 (Fig. 3). This is interpreted as the expression of the thrusts, resulting from the limited Alpine convergence the led to the partial closure of the Bay of Biscay (Alvarez-Marron et al., 1997; Ayarza et al., 2004; Fernández-Viejo et al., 2012). Some of the focal mechanism in this area give a component of inverse fault, which would agree with this interpretation.

b). **Events offshore Lisbon, WIM (South)**. This second band runs along 670 km between the coast of Portugal and 16.8ºW. We can subdivide two subclusters, the first one oriented N80ºW from the coast until 12.5ºW. 320 km apart, a small bend occurs adopting a position N75ºE along a further 160 km until 16.8W. This band presents more events than the Galicia one. In this sector, for the 20-year period analysed, we got a total of 295 events with magnitude over 2.5, (15 events/yr.), 20 over 4-4.9, (1/yr.) and 3 over 5, (2 in the last decade). Most of them are concentrated in the first 30 km of depth, being progressively more superficial, as crust thickness diminishes oceanwards. The β-value is 0.96.

The focal mechanisms of the far away events than 14ºW present a significant component of strike slip. The depth of the events, which relates to an unknown error, is more than the Moho depth, which in this area decreases from 23 km in the coast to 12 km oceanwards, at the end of the lineation.

Again, in the depth distribution profiles (Fig. 3, profile 2) it seems that there is more events up to the transition between hyperextended crust and exhumed mantle and a lack of them in the oceanic crust until the western termination where there is an increase in the amount of seismicity and at greater depths.

There is a puzzling vertical alignment (Fig. 3, profile 5) below the Extremadura Spur, indicating some type of structure northwards and probably a second one within it. There may be a manifestation of a splay fault delimiting this topographic high.

A detail worth to note is that the highest magnitude earthquakes in this area occur in the subcrustal mantle and below two seamounts (Fig. 3, profile 4) that may correspond to volcanic edifices (Escada et al., 2022). The referred vertical alignment would be consistent with a volcanic origin for those particular events.

Although the graphics show how the depth of the earthquakes is accordingly diminishing towards the ocean along the whole WIM, there is a considerable amount of earthquakes nucleating in the uppermost mantle, and especially in the transition areas to exhumed mantle domain and towards the oceanic crust at the end of the alignments. This suggest that seismicity occurs in the whole margin crust, down to lithospheric levels. Within the Lisbon (S) alignment, the hypocenters are slightly deeper than in the Galicia (N) alignment, even when crustal depths are similar in both areas. The dispersion of events is greater at around longitude 14ºW, maybe indicating the presence of some type of seismic boundary or discontinuity in the N-S direction that traverses the whole WIM. In general, earthquakes along both alignments seem to cluster around marine reliefs, of either tectonic origin (Galicia Bank) or with volcanic additions (Extremadura Spur).

Earthquakes located below the Moho may indicate a strong, seismogenic upper mantle that can sustain large stresses, later released during brittle rupture. This type of "ductile earthquakes" have been proposed for Wyoming lithospheric mantle (Prieto et al., 2017), or in Newport-Inglewood fault attributed to a system of seismic asperities in ductile fault zones. Once a ductile mylonitic structure has developed a shear zone, subsequent cataclastic deformation is consistently localized in a narrow zone (Takahashi et al., 2017). The role of temperature in the rheology of the oceanic lower crust and lithospheric mantle is not well understood. Earthquakes occur throughout the oceanic crust and upper mantle, the later with olivine-dominated rheology. The brittle ductile transition is associated with a threshold temperature of about 600ºC, which represents a transition from velocity weakening to velocity strengthening, consistent with the focal depth of earthquakes in the oceanic lithosphere elsewhere (Boettcher et al., 2007).

In the SWIM, Grevemeyer et al. (2016) provide evidence for earthquakes rupturing at 30-50 km depth, hypothesizing that they are caused by the proximity to the AGFZ plate boundary and by an elastic behaviour at the continent-ocean transition zone, which is further supported by low heat flow and the amount of regional stresses caused by the Africa and Eurasia? collision. Being and old passive margin, shortening and deformation of such a rigid lithosphere, may cause intense deformation at mantle depths. Mantle nucleation of events may additionally be encouraged by the presence of fluids or fluid migration in the mantle,

either by reducing the effective normal stress or promoting strain localization along the shear zones. The velocity gradients observed on the edges of rift domains would make them preferred sites for reactivation due to inherited strength contrasts down to mantle depths.

The focal mechanisms in the study area, corresponding to events of magnitude 4.1 to 5.9 (Table 2), indicate mostly strike-slip movement.

### 3.2 WIM seismicity and geophysical data

The superposition of seismicity onto different types of geophysical data can be used to infer relationships that may give hints to the seismicity sources. In the absence of confidently mapped submarine faults in the area, figure 4 depicts the event distribution obtained in this study overlaid to a different set of geophysical data for assessment. In the first case, the events are overlaid onto the Moho depth map. As expected, depth of events follows crustal thickness decrease seawards. However, as many events seem to nucleate in the uppermost mantle, we must infer that the crust and the upper mantle in the WIM must be strong, and that neither the Moho discontinuity nor the transition upper-lower crust are the rheological fundamental boundaries concerning seismicity.

When the seismicity is overlaid on the calculated gravity (Fig. 4b), the main relation arise from the fact that seismicity indeed does align where the basement is uplifted (i.e. Galicia Bank or Extremadura Spur), and therefore, follows the gravimetric expression of thickened crustal blocks, something that could be expected given that gradients of velocity influence on crustal strength. Regarding the magnetic map (Fig. 4c), it is difficult to stablish a relationship, attending to the main N-S disposition of the magnetic anomalies. Finally, figure 4d shows the seismicity overlaid onto the WIM rift domains map, which shows that seismicity does not nucleate preferentially in any domain. Although events are somewhat more abundant in the proximal domain, they do appear on exhumed mantle and hyper-thinned crust, indicating that the weakness of the domains is adequate for producing earthquakes anywhere. Seismicity almost abruptly stops around the area of undisputed oceanic crust. Nonetheless, there is still a few events westward toward the MAR, and they follow the N80ºW direction too.

Moreover, it also shows that the south alignment happens south of the boundary between Northern and southern WIM, if we consider this boundary the location of the change in width of the exhumed domain, therefore ignoring rift domains altogether.

## 4 Causes of the intraplate seismicity along the West Iberian margin

The concentration of seismicity at the northern and western coasts is explained in general terms as the internal deformation being the prime mode to accommodate strain induced by the convergence between Africa and Eurasia and the impossibility of propagating west of the Iberian peninsula due to the Atlantic ridge push.

Previously to discuss the possible sources of the WIM seismicity, a summary of the characteristics of the studied events that are reliable and therefore treated as strong constraints are:

a) Seismic events are concentrated along two distinct WNW-ESE trends parallel to the active AGFZ Eurasia-Africa plate boundary. The interval distance between bands is similar, around 300 km (Fig. 2).

b) Magnitude is low to moderate, and hypocentral depths include the whole crust and upper mantle.

c) The bands do not follow inherited tectonic trends. The events cut different rift domains and are prone to cluster along transitional areas and topographic highs. There is not an evident link to gravity or magnetic anomalies.

d) They do show an association with topographical highs, especially with the western border of the Galicia Bank and the northern part of the Extremadura Spur.

e) The few focal mechanism available indicate overwhelmingly strike slip types.

A note on reliability of observations and data limitations

The text highlights the limitations of seismotectonic interpretations due to the inconclusive mapping of structures in the sea floor of the WIM. Despite being well probed, the margin contains many highs, roughness, and bathymetric heterogeneities that make it difficult to map tectonic features, specifically faults that could act as seismogenic sources. The structural variability observed along the margin added uncertainty in the data processing when introducing velocity models in the location programs of the events. Additionally, crustal depths in the margin vary, but a considerable number of events show hypocentres below the interpreted Moho depth.

The acquisition mode poses the second limitation as the land stations are located to the east, leading to a biased location error. This results in uncertainty in some locations and the lack of full azimuth to relocate with confidence. To minimize the impact of these shortcomings, only the events that comply with the requirements were considered reliable and retrieved:

a) Magnitude should be over > 2.5.

b) Events with location uncertainty of < 25 km in plant with average values < 10 km, and 75 km in depth. It is precisely this last parameter the one that poses the main problem, although the average error estimated has been of 4.9 km.

c) The average GAP < 183, for all the events used in this study being the ones with higher GAP discarded for this purpose.

Based on the aforementioned observations, we may have a walk through the various hypotheses that have been proposed to explain seismic activity in the WIM. We can evaluate the validity of these hypotheses against the data collected over the past two decades.

**4.1 Intraplate seismicity generated by unmapped faults**. Strike-slip corridors, shear zones, transform faults.

Some causes for local clusters of events within the WIM have been put forward in the literature: for instance, Vazquez et al. (2008) indicated that the GB resulted from the reactivation of two major faults, forming an ample shear zone. Borges et al. (2001) related the central WIM seismicity to strike-slip movements in certain structures. Recently, Somoza et al. (2021) indicated that to the north of Nazaré fault, seismicity responds to a reactivation and inversion of thrusts situated in the ocean seamount, that even connect to faults in the continent. The Nazaré fault would be prolongated up to the MAR, linking with the Kurchatov transform fault (Fig. 1). However, there is not enough data to support this interpretation.

In any case, local clusters do not explain the overall distribution in the WIM. We have observed also local clusters such as the southward directed in the GB related to the arrested subduction in the Bay of Biscay or the vertical alignment beneath some volcanic edifices along the Extremadura Spur (Figure3, profile 5).

For the bigger picture and an inclusive theory for the WIM seismicity, one of the recurring hypothesis indicated by some authors was the presence of shear corridors, (Custodio et al., 2015; Whitsmarsh et al., 1990) due to the particular stress state of the margin and the Iberian Peninsula.

Their oblivious indifference to structures of the crust or rheological boundaries of the rifted margin, all support the view that the events we are studying respond to the actual stress state, drawing the incipient silhouette of rectilinear lithospheric fractures or zones of strain parallel to the south and north well-formed transform faults, one of them a plate boundary. These events may signal the formation of fracture zones that mimic oceanic transform faults. Contrastingly, to the ones formed during the ocean opening, these ones are forming in the opposite direction, from land to sea. These faults may correspond to accommodation structures of the extension and opening of the Atlantic in these latitudes, where oblique extension and the jump of plate boundaries (Srivastava et al., 1990), might have impeded the formation of classic oceanic transform faults.

This process would be framed within the peripatetic behaviours of the Iberian microplate. The topic of oblique spreading along the WIM has not been deeply studied. However, there is some evidence that the opening of the Atlantic, and especially during the Bay of Biscay rift, Iberia, may have been "bouncing" from its perpendicular to the extension direction. Favouring this option is the fact that oblique spreading is very common in slow spreading ridges, as lithosphere is relatively cold and crustal growth is decelerated (Peyve, 2009). Another argument in agreement with this hypothesis is that mid-ocean ridges with oblique spreading are not dissected by transform faults. Shallow tectonics dictates their segmentation rather than mantle convection.

In the other hand, the presence of these heterogeneities stand out from the rift domains map, where the width of domains changes at the Lisbon lineation and north of the Galicia lineation (Fig. 3).

Published interpretation relying on the study of seismic profiles that cross the WIM seismicity bands, do not show evident crustal strike-slip structures. However, it is important to note that sub-vertical strike slip faults that promote horizontal displacements are difficult to image in near-vertical reflection data, and have a limited expression on the floor surface, even more if they are relatively recent.

The claim for these NW-SE oriented transform corridors to facilitate extension along the MAR, working as "pseudo transform faults" (Fig. 5a), reinforces the need to accommodate differential stresses from South to North along the WIM, which have resulted from the rotations and subtle drift of the Iberian plate during the past 30 Myr, together with the current NW-SE oriented compression. In this case, the seismicity alignments may correspond to those shallow accommodation zones, similar to shear zones that act as nascent oceanic transform faults, without dissecting the ridge (yet).

This compartmentalization of a passive margin into strike slip narrow belts between the MAR and a deformable plate interior, is more consistent with soft plate idealizations than with perfectly rigid plates and narrow boundaries (Storti et al., 2007), which takes us to the next hypothesis to try to fit this seismicity into the plate tectonics theory.

### 4.2 A **Diffuse and ~1000 km wide plate boundary**

Plate rigidity and narrow plate boundaries are central assumptions of the original plate tectonics theory. But in the last decades evidences have been piling up to suggest that in many cases, plate boundaries can be very wide and plates may not be as rigid as assumed (Ribeiro, 2002). The diffuse plate boundary hypothesis try to reconcile observations related to intraplate activity proposing that large plates may be composed of a few smaller subplates and crustal blocks, so more rigid portions are in relative motion with each other separated by diffuse boundaries with clustered deformation (Gordon, 1998). The diffuse plate boundaries are much broader than traditional boundaries such as mid-ocean ridges, trenches and/or oceanic transform faults (Zatman et al., 2000).

In this way, the passive interior of tectonic plates can be dissected by strike slip deformation belts propagating towards MAR segments that separate adjacent, partially coupled lithospheric slivers with differential translations, and deformation rates. These internal differential displacements are then accommodated by distributed deformation within the plate interior, so that localized deformation is ultimately dissipated into a non-rigid plate.

Applying this model to WIM is challenging because

published studies proposing soft plate deformation and the formation of weakened zones involve the spatial extension of transform faults through oceanic fracture zones, possibly weakened by serpentinization, that are used to transfer deformation from mid ocean ridges towards neighbouring plate interiors. In the WIM case, as pointed out earlier, the Galicia (N) and Lisbon (S) lineation are discontinuous, do not intersect the MAR and specifically stop in the vicinity of the oceanic crust, so a full lithospheric detachment is not occurring. Therefore it is more likely that these lineations may be inherited from the rifting period, or from post rift accommodation and thermal subsidence rather than to the sea floor spreading stage of the Atlantic.

Along the southern Biscay margin, the Ventaniella seismicity line (Fernandez-Viejo et al., 2014; López-Fernández et al., 2018) shows a similar NW-SE trend and includes earthquakes of similar magnitude, distribution and focal mechanism as the Galicia (N) and Lisbon (S) lineation identified in the WIM. Contrastingly, it appears in a continental margin whose direction is perpendicular to the WIM direction (the Bay of Biscay margin). If the three seismicity bands are expressions of shear corridors inside an exceptionally wide plate boundary, this hypothesis would deserve its credit, and the fact that they appear in two completely perpendicular trending passive margins, even crossing oceanic and continental parts of the Iberian plate, would agree with it.

Asti et al. (2022) proposed a 400 km wide, diffuse Iberia-Eurasia plate boundary transecting Iberia during the Mesozoic from the Iberian Rift system to the Armorican shelf, which was associated with the contemporaneous opening of the North Atlantic and Bay of Biscay. The deforming domain between these two boundaries was a fuzzy region, where the exact locations of different tectonic structures and rift basins is still debated. If a similar diffuse plate boundary may exist today, it would be almost double in width and would encompass the entire Peninsula except for its northeastern third.

One advantage of the diffuse plate boundary theory (Fig. 5b) is that some of the inconsistencies between the deformations inferred from plate kinematic models and geological observations (which are extraordinarily abundant in Iberian kinematics- see; Barnett-Moore et al., 2016, and references therein) may be then reconciled. The same amount of total displacement is

partitioned between a series of subparallel strike slip corridors, each of them accommodating a portion of the total displacement between the stable parts of the African and European plates. This would significantly broaden the range of geodynamical models in a region with such as slow lithospheric deformation. If deformation between Africa and-Eurasia is distributed inside Iberia, these internal corridors of deformation can accommodate the convergence of both macroplates, while assisting the E-W Atlantic extension.

Nevertheless, if a great part of Iberia belongs to a "diffuse plate boundary", we still need to explain why the corridors are produced at those particular separation of about 300 km. Does that separation depends on the width of the subplates related to the "rigid microplate"? Does it depend on the velocity rate of the ridge? This needs further investigations out of the scope of this study, but it points to some kind of relationship to be able to form measurable strike-slip corridors between areas with contrasting crustal lithospheric thickness, plate width and undergoing different stress fields. In any case, a complete radical hypothesis has been advanced in the literature, which approaches the dynamics of the WIM in a very different perspective: the closing of the Atlantic Basin.

### 4.3 **Subduction initiation and oceanic dynamics that may be, or have been, at play in the WIM**

Several authors have suggested the possibility of a developing subduction zone in the SWIM, which could explain the moderate seismicity and the large magnitude of the 1755 Lisbon earthquake in an intraplate setting (Vilanova et al., 2003; Martinez-Loriente et al., 2021). Ribeiro et al. (1996) proposed that the WIM is the site of an incipient northward propagating subduction zone nucleating at the Gorringe Bank (figure 5c). The ongoing terminal stage of collision between Africa and Eurasia produces compressive stresses along the SWIM, which could trigger passive margin reactivation. NE-SW thrust systems extending 300 km along the WIM accommodate the arc-orthogonal convergence (Gutscher et al., 2012) and younger thrust faults are nucleating along the west Portuguese passive margin or in the Tagus Abyssal plain. Duarte et al, (2013) suggest that these structures may be indicative of the onset of margin tectonic inversion and the nucleation of a new subduction.

However, the buoyancy of old and cold oceanic lithosphere makes the spontaneous initiation of subduction in passive margins difficult, and significant weakening mechanisms are required to fail in subduction (e.g., Stern and Gerya, 2018). The Azores, Madeira plume, or the peridotite ridge in the Galicia margin may serve as potential weakening mechanisms. A 100 my old oceanic crust that would need a significant compressive stress to fail (Zhong et al., 2019), which is not observed at the WIM. Thermo-mechanical models suggest that magma-poor margins with hyperextended and exhumed mantle domains are favourable sites for subduction initiation, as the serpentinized mantle facilitates strain localization and the progressive development of a major shear zone that ultimately links up with a zone of high strain the continental lower crust along the necking zone (Auzemery et al., 2021). Shearing then propagates into the mantle or initiate a subduction plate boundary. The exhumation and hydration of the mantle in the footwall of major detachments induces weak decollement layers (Brun and Beslier, 1996) prone to tectonic reactivation, which would diminish the required compressive stress to initiate subduction (Hirt et al., 2013). Subduction could be easier in the southern WIM as a relevant transform fault, the Gloria fault, puts in contact two different lithospheric age crusts making for additional stresses and weakened lithosphere. Although the initiation

of subduction by passive margin collapse is extremely unrealistic, i.e. (Stern and Gerya., 2018), the westward migration and propagation of the Gibraltar Arc along the SWIM may be responsible for induced subduction initiation (Duarte et al., 2013). The majority of focal mechanisms along the WIM events outside of the plate boundary are strike-slip, which does not support the lithospheric collapse in a subduction interface propagating northwards along the WIM. Although some type of underthrusting cannot be ruled out, seismicity does not support unequivocally the presence of a subduction zone.

### 4.4 **Oceanic complexity, inheritance, formation of strained corridors in the Atlantic ocean**

Within the other big oceanic basin of the Earth, in the Pacific, crustal studies evidence complex tectonic settings. In triple junctions, triangular microplates without any bounding continental margins grow, as exemplified by the Galapagos microplate. Researchers have found strong evidence that indeed, one of such microplates grew to become the Pacific plate, for instance (Boschman and Hinsberger, 2016). Schouten et al. (2008), studying distributed deformation at oceanic triple junction showed that lithospheric plates undergo significant internal deformation as their boundaries rapidly evolve, as exemplified by the RRR Cocos-Nazca. The Cocos Nazca rift tip does not meet the East pacific rise. Instead, two secondary rifts form the link. The active incipient rift is just the latest of a sequence of southeast trending fractures that progressively stepped during the last 5 Myr, successively accommodating minor N-S extension of the generated oceanic crust (Smith, 2011).

The roughness of the abyssal plain around the King´s Through may represent the incipient and left-alone rotation of a potential mini Galapagos-like microplate. The current scars related to the triple junction on the King´s Through area with seafloor expression relate to inherited lithospheric fractures, which developed during the evolution of the triple point and the plate boundary migration. However, even when considering the inheritance of such lithospheric discontinuities, the seismicity in the WIM does not appear connected to it in particular, more than following the general NNW-SSE trend. And as figure 2 shows, seismicity is mainly within the continental rifted crust and not so abundant within the oceanic domains.

Palano et al. (2015) propose a current large-scale clockwise rotation that makes Iberia to act as a microplate with a southern limit in the AGFZ Nubia-Eurasia convergent zone. Somoza et al. (2021) indicate that the main weakened zones along the N and NW Iberia are the old inherited limits between the oceanic domains of Eurasia and Iberia (King´s Trough, Azores-Biscay rise, Jean-Charcot Seamount). They question the clockwise rotation in the oceanic domain, which would be dominated by the stress derived from the ridge propagation in the segment between Azores and King´s Trough, with a N120º propagation direction. They also suggest a left-lateral shear zone resulting from the northward movement of the continental Iberia and the SE movement of oceanic Iberia along the OCT.

According to the possible hypothesis above mentioned, the one that presents less problems in this study is that the pre-existing Mesozoic rift structure, together with the oblique oceanic spreading, when the plate boundary was shifting southwards towards the AGFZ, may have left weakened areas along the WIM. These inherited zones may be prone to reactivation and release seismicity under the actual stress regime. Inherited features revealed by the seismicity are for instance, the old proto-subduction interface in the north of Galicia Bank (figure 3, profile 5) or the associated volcanic edifices around the Extremadura Spur.

## 5 Conclusions

Analysis of two decades of seismicity data recorded by the Spanish and Portuguese seismic networks along the West Iberian passive margin (WIM), covering an area between 38º and 45ºN latitude, shows that two NNW-SSE bands concentrate 98% of the intraplate activity. The northern band is located offshore Galicia, while the southern band is in offshore Lisbon. The observed seismicity is higher than expected for a passive margin, and coincides with the area where the mid-Atlantic ridge lacks conspicuous transform faults. Focal mechanisms indicate strike-slip deformation, and hypocentral depths suggest crustal and upper mantle sources.

Various hypotheses have been proposed to explain this seismicity, including the formation of shear corridors/transform faults, a diffuse plate boundary, initiation of subduction, and other less understood mechanisms associated with weakened zones inherited from the Mesozoic rifting, complex ocean opening, and/or triple point migration in the Iberia microplate. Based on the parameters obtained in this study, a conservative approach to explain the WIM seismicity would be the presence of strained shear corridors that nucleate preferentially on topographical heterogeneities of the seafloor. These corridors form in response to the accommodation of two perpendicular sources of stress: the E-W extension associated with the MAR, and the N-S compression from the Azores-Gibraltar Fracture Zone. The drastic crustal thickness variations from east to west and the nucleation of small events along multiple faults that make up the block morphology of the Mesozoic rifted margin conform the occurrence of areas prone to strain release. The presence of sub-Moho earthquakes also indicates that the upper mantle in the WIM acts as a high-strength layer, particularly beneath the hyperextended and exhumed mantle transition areas.

Several plate-scale processes may be acting simultaneously to produce the observed seismicity, such as the possible initiation of convergence-related deformation in the SWIM or the inherited compartmentalization of the lithosphere in the northern part of the margin due to triple junction migration. Further study of seismicity in the WIM and neighbouring oceanic areas, together with the arrival of seafloor stations, may provide further insights to better understand this marine intraplate clustered activity.

## Data and resources

Seismicity data used in this study were collected from the IGN Spanish seismic network at www.ign.es (last accessed December 2022) and from IPMA Portuguese seismic network at www.ipma.pt/en/ (last accessed December 2022). Gravity, magnetic and crustal data were obtained from GeoMapApp at https://www.geomapapp.org/ (last accessed January 2023).

## Acknowledgements

The research leading to these results received funding from the Project MCIU-22-PID2021-123116NB-I00 by the Spanish Science and Innovation Ministry. P. Cadenas hold a Marie Sklodowska-Curie Actions, SUBIMAP, grant agreement ID: 895895. Chat GPT was used to improve the English writing and clarity in some paragraphs.

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

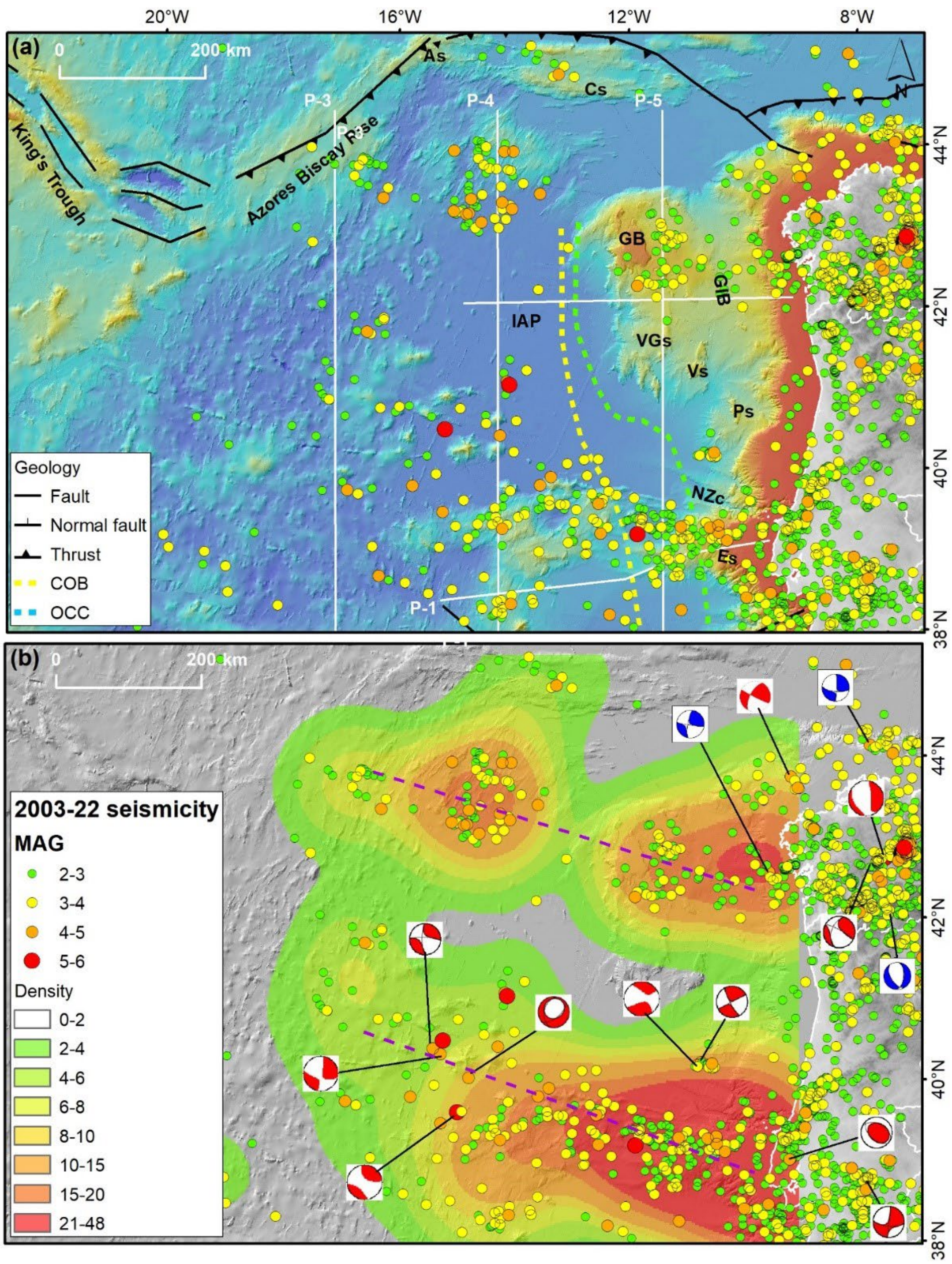

755

**Figure 2**. a) Seismicity studied in this work showing the events according to magnitude. b) Density map of events studied in this work highlighting the alignments in purple discontinuous lines 1) Galicia, north and 2), Lisbon, south (see text) some focal mechanisms are shown. Seismicity catalogue in the WIM in the period 2003-2022 (magnitude > 2.5, RMS < 1, Smaj < 25, Smin < 25 and Err < 75), according to the parameters described in the methodology section. Topographic base from EMODnet Bathymetry Consortium (2020). Focal mechanism from CMT Catalog and IGN Seismic Moment Tensor Database. P1 to P5 white solid lines show the position of the profiles shown in figure 4.

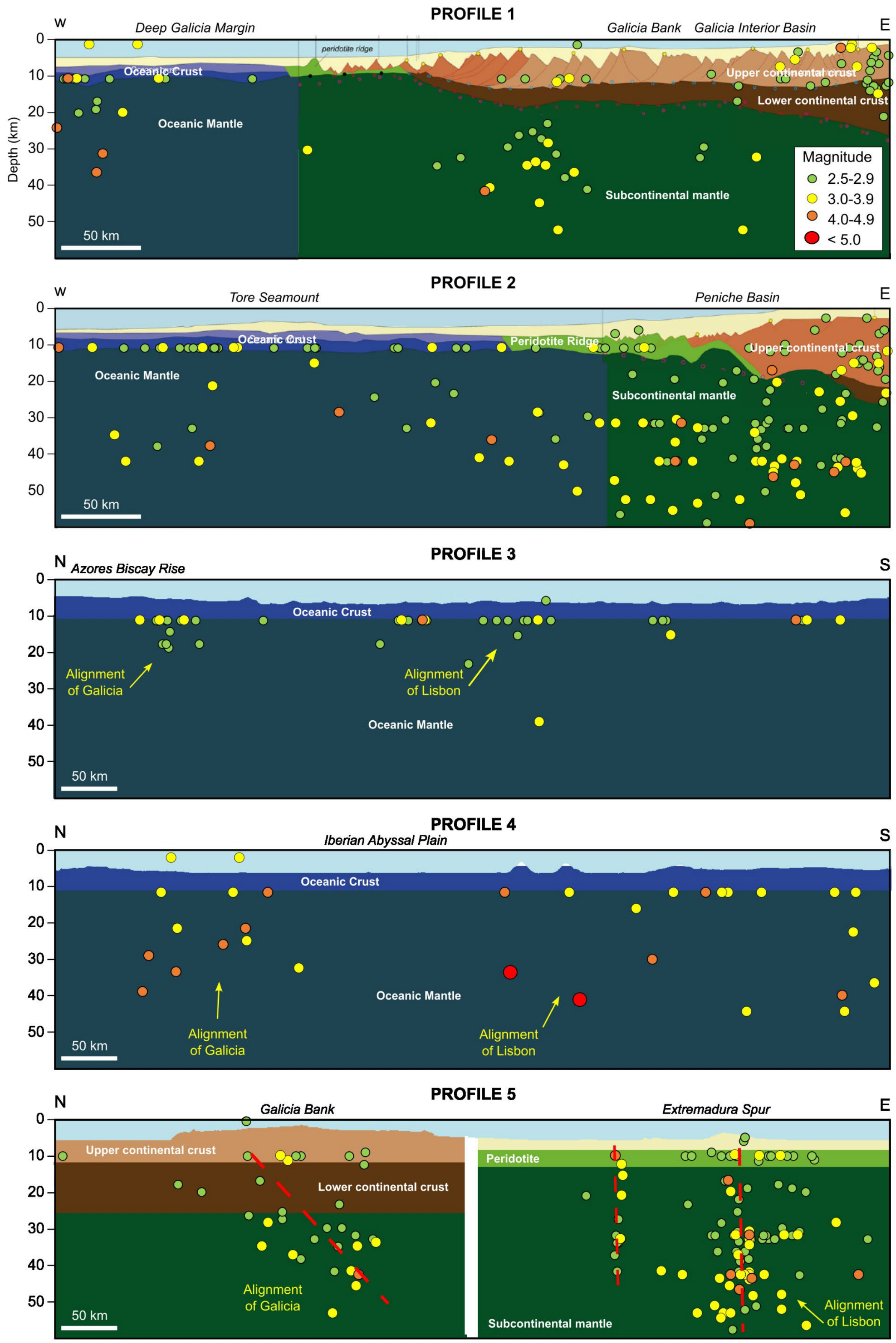

Figure 3. Representation of cross-sections of seismicity at depth overlaid on crustal cross-sections modified from Granado et al. 2021, showing the crustal domains. The location of the profiles is shown in figure 2a and the seismicity included in each profile comprises 100 km band along that direction. Profile 1) Depth profile of seismicity along alignment North (Galicia) b) Profile along alignment south c), d) North south cross-sections along different longitudes, oceanic domain, and e) close up in cross -section of the seismicity clusters around Galicia Bank and Extremadura Spur.

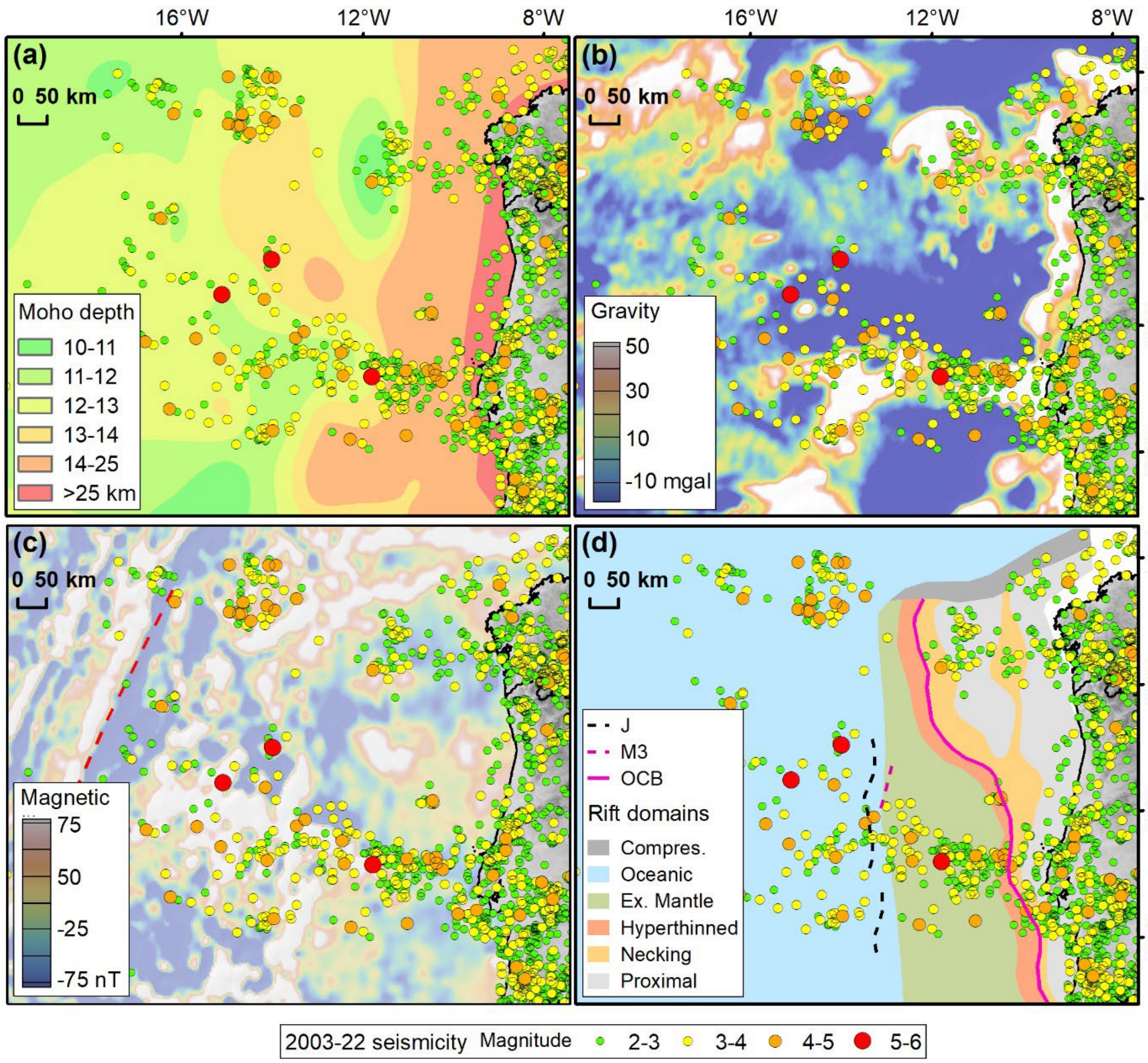

**Figure 4**. Map of the WIM seismicity located in this study overlaid on: a) crustal depth of the WIM (Laske et al., 2013); b) gravity map (Sandwell et al., 2014); c) magnetic map (Meyer et al., 2017); d) rift domains of the WIM (Granado et al., 2021).

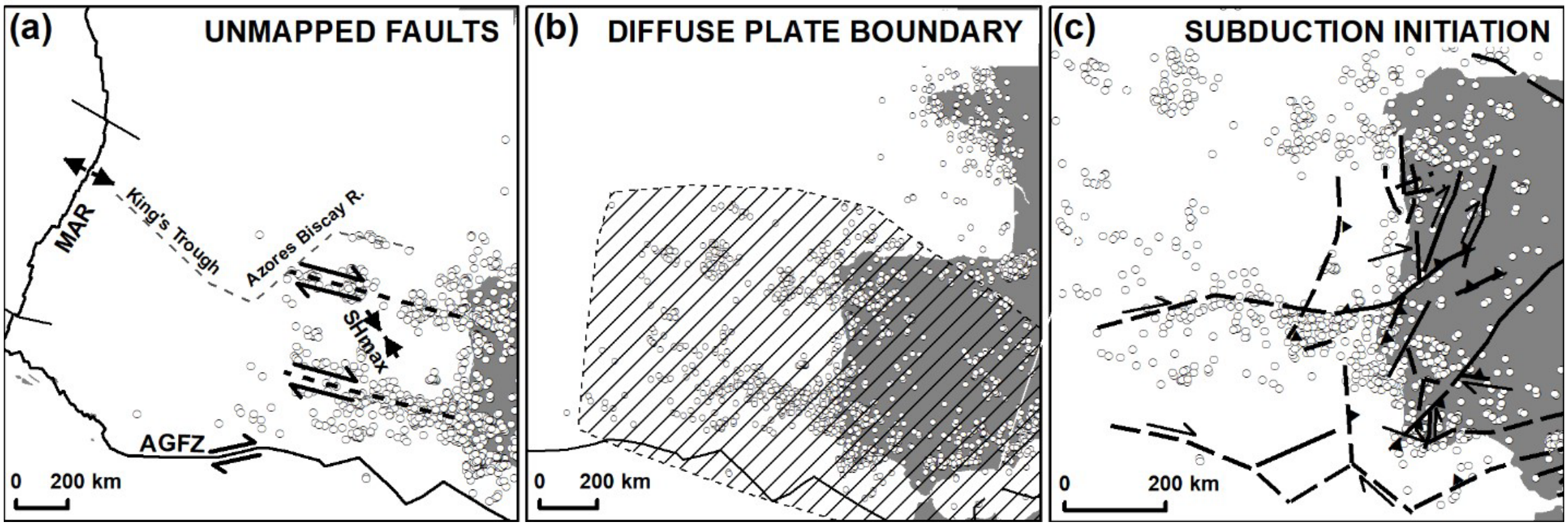

**Figure 5**. Schematic cartoons to illustrate the different hypothesis on the origin of seismicity of the WIM. a) shear corridors, inverse transform faults b) diffuse plate boundary c) initiation of subduction.

795

## TABLES

**Table 1**. Synthesis of the statistical parameters of the seismicity catalogue prepared from the data of the IGN and IPMA seismic networks.

| | RMS | Smaj | Smin | Err | GAP |
|---|---|---|---|---|---|
| Minimum | 0.10 | 1.5 | 0.8 | 0.1 | 4 |
| Maximum | 0.99 | 24.8 | 22.9 | 74 | 335 |
| Average | 0.57 | 9.48 | 5.56 | 4.98 | 182.44 |
| Standar deviation | 0.21 | 5.59 | 4.09 | 11.46 | 87.29 |

800

**Table 2**. Focal mechanisms collected for the study area from CMT Catalog and IGN Seismic Moment Tensor Database.

| LONGITUDE | LATITUDE | DEPTH (km) | Mw | DATE | DATABASE |
|---|---|---|---|---|---|
| -14.4400 | 39.4800 | 32 | 5.9 | 24/01/1983 | CMT |
| -09.1204 | 43.7570 | 27 | 4.3 | 23/04/2006 | IGN |
| -14.2965 | 39.9127 | 3 | 4.3 | 18/08/2007 | IGN |
| -10.6200 | 40.1800 | 19 | 4.8 | 04/09/2018 | CMT |
| -10.5665 | 40.2162 | 29 | 4.9 | 04/09/2018 | IGN |
| -14.7700 | 40.1800 | 12 | 4.7 | 01/11/2018 | CMT |
| -14.9012 | 40.2458 | 25 | 4.7 | 01/11/2018 | IGN |
| -07.7683 | 44.1478 | 21 | 4.2 | 29/10/2021 | IGN |
| -09.4919 | 42.5781 | 12 | 4.1 | 27/01/2022 | IGN |