# Peer review of "Offshore seismicity clusters in the West Iberian Margin illustrated by two decades of events."

_EGUsphere, 2023_

## Referee Comment (RC1)

**Offshore seismicity clusters in the West Iberian Margin illustrated by two decades of events.**

By Gabriela Fernandez-Viejo, Carlos Lopez-Fernandez, Patricia Cadenas

**Comments:**

L45-47: If the "W-E thrusts" refers to the Gorringe Bank, change it to "NE-SW thrusts"

L59: The Africa-Europe plate boundary continues east of the Strait of Gibraltar. I suggest to rephrase the expression

L62: Add space "theWIM"

L62: what is the "WIM abyssal plain"? If it exists, locate it in figure 1.

L63: Add "." after ")"

L70: Ramos et al., 2016 focused on the Algarve basin, southern Portugal… nothing related to structural domains along the WIM.

L78-80: Sallarès et al., 2013 and Martínez-loriente et al., 2014 presented geophysical evidence suggesting that the Gorringe Bank and the neighbouring abyssal plains are composed mainly of exhumed mantle rocks and presented a new model for the opening of the North Atlantic. I suggest including these references.

L87-114: I highly recommend adding a map with the domains referenced in this section, and the delimitation of the different segments of the WIM. It is very difficult to follow the (messy) description of the authors. Therefore, I also suggest rewriting the section.

L90-92 & Fig 2: "The OCT extends between 12º10'W and 12º30'W in the IAP (Whitsmarsh et al., 1990) and it would extend Nº10 for 130 km until Extremadura Spur (…)" It is very difficult to correctly locate the COT with the map coordinates. I highly recommend adding more subdivisions between coordinates. How can readers locate longitude 12º10'W if all the information they have is the location of longitudes 12ºW and 16ºW?

L91: 10ºN, Sure?

L94: "The exhumed mantle domain has been drilled at a serpentinite ridge (Boillot et al., 1988b)" Where?

L95-98. Add reference

L100: DGM? It is not located on the map and the abbreviation is not described

L102: "The GB is 15-20 km thick"… and? Made of?...

L107: What is "THD" y "dZ"? where are these maps?

L111-114: Let's see, if the WIM is divided into 3 segments, and the authors say that there is a segment further south than what they call "South WIM", wouldn't it be more logical to call it Central WIM or something similar?

L116-120: when? it should be specified to which period the authors refer

L135-136: when? At present? At the beginning? Has it been constant over time?

L138-142: I suggest the authors include some arrows in figure 1 that indicate the kinematics along the plate boundary.

L142: Again, Ramos et al. only investigate a small portion of the SW Iberian Margin, the Algarve basin south of Portugal. There are many other works that propose the reactivation of thrusts throughout SW Iberia at a regional level, such as Martinez-Loriente et al. 2013 (but there are many others). Ramos's work is very local and their conclusions quite debatable.

L144: The Gloria Fault is the source of one of the largest earthquakes occurred in the North Atlantic, 1941 Mw 8.3-8.4 (e.g., Baptista et al., 2016)

L146-152: In this section the authors mix nanostrain/yr and mm/yr... for non-experts, it is difficult to compare the different geodetic velocities.

L179: Why have you included historical seismicity but not the instrumental seismicity available prior to 2003? What's the point of including the first and ignoring the second? I would like to know what these 10 historical earthquakes are, and if there is any relevant aspect, that it be recorded in the figures, text, in a table (somehow).

L182: "The 9 focal mechanisms considered in this work have been obtained from the CMT Catalogue". Figure 2 includes 14 focal mechanisms (not 9 as mentioned by the authors)

L185: "(ii) geological structures mapped in the continental platform (Somoza et al., 2021)". Why only those included in the continental shelf?

L195: "Further up towards the MAR, there is some isolated events.". Could this lack of seismicity be associated with the distance to the onshore stations?

L196: "The orientation of these bands is about N80ºW". In my opinion, the southern alignment has a clear E-W orientation and it is related to the Estremadura Spur and the Tore Seamount (it is not located on figure 1).

L202: "This density band of events is oriented N75ºW from the coast until 17.5ºW". I disagree with the interpretation of this alignment. There is an E-W alignment from the coast until 12ºN related to the Galicia Bank and a second cluster of seismicity to the northwest related to another ridge/ relief (without name in Figure 1). Between both clusters of seismicity there is a gap of more than 100 km without seismicity, so there is no evidence to indicate a relationship between them.

L211: "The number of events is larger at the transition between hyperextended crust and exhumed mantle" As it is not indicated in figure 3, I do not know what the authors consider to be the hyperextended crust and the exhumed mantle domains. According to my consideration (which coincides with that of Granados et al., where the profiles come from), there is exactly 1 earthquake in this segment (Profile 1). Therefore, I think the authors' statement is wrong.

L211: "There is an arguably but noticeable 50 km wide gap in event distribution west of the Galicia Bank". Why this gap is "arguably"? In 150 km there are exactly 2 earthquakes.

L2014-2015: "especially within the transition between the hyperextended and exhumed mantle domains". Same as in the comment of Line 211.

L215-216: "A particular set of south-dipping earthquakes can be observed in profile 5". This is highly debatable. The seismicity could be vertically aligned, or even dip to the southwest but with a lower dip than that interpreted in figure 3 by the authors.

L218: "Some of the focal mechanism in this area…". There is a lot of distance between the few focal mechanisms shown in figure 2 and this seismicity. By the way, why are some focal

mechanisms represented in red and others in blue? It is not indicated in the legend or in the figure caption.

L219-221: As I mentioned before, I see this seismicity aligned E-W from the coast up to 14 or 15ºN (? = it is difficult for me to be precise with the low coordinate discretization of figure 2)

228-230. In this case, I agree that there is an amount of seismicity in the transition between hyperextended crust and exhumed mantle, but I disagree with "a lack of them in the oceanic crust until the western termination…" the seismicity decreases, but there are 15 or 20 earthquakes.

L231: It would be interesting to know which segment of profile 5 of Figure 2 is represented in Figure 3, since it would allow to locate on the map these possible vertical seismicity alignments.

L234-235: "… the highest magnitude earthquakes in this area occur in the subcrustal mantle and below two seamounts…" What??? In profile 4 the seismicity is projected (100 km). If we look at the map (figure 2), these 2 earthquakes that the authors refer to are located far from these two seamounts or volcanic edifices.

235-236: "The referred vertical alignment would be consistent with a volcanic origin for those particular events." Are the authors referring to the vertical alignment mentioned in the previous paragraph (231-233)? If so, it is difficult for me to understand the relationship that the authors see between this seismicity of the Estremadura spurn that is seen in the southern part of profile 5 with these two earthquakes that are seen in profile 4 and that the authors say are related to two volcanic edifices (which actually aren't)?

L254-256: Geissler et al. 2010 already showed that in SW Iberia the majority of seismicity occurred between 40-60 km depth, and with strike-slip or inverse focal mechanism solutions. Bartolomé et al. (2012) associated the strike-slip seismicity with the Lineament North and Lineament South strike-slip faults. Martínez-Loriente et al. (2021) associated the deep inverse seismicity as well as the largest seismic events occurred in the region with the HAT.

L256: "?" Delete it

279: "Seismicity almost abruptly stops around the area of undisputed oceanic crust. Nonetheless, there is still a few events westward toward the MAR, and they follow the N80ºW direction too.". In the north, the seismicity stops just before the COB (around 12ºW), more than 50 km before the oceanic crust. In the south, seismicity does not stop at any point and continues from one domain to another

L290: in figure 2 it does not include the AGFZ, so I cannot get a visual idea of what the distance is between it and the southern alignment.

L310: "GAP"? Describe the abbreviation

L343-346: There are a lot of scientific publications showing strike-slip faults with MCS data. I can include 10 or 15 references only in the SW of Iberia. I know that the authors have access to seismic profiles acquired in the WIM. If they don't see the strike-slip faults, could it be that these structures don't really exist?

L408: "NE-SW thrust systems extending 300 km along the WIM accommodate the arc-orthogonal convergence (Gutscher et al., 2012)". What are these fault systems??? Specify them and add references. Gutscher et al., investigated the possible subduction under the Gibraltar Arc, nothing related to the WIM or any "thrust system" there.

L409: "and younger thrust faults are nucleating along the west Portuguese passive margin or in the Tagus Abyssal plain". Which ones? Specify them and add references where the existence of

these structures can be verified. The work of Duarte et al. (2013) does not count as a reference since they only presented a theory without a single real data to support it.

L416-428: I am surprised that the authors do not consider the Gorringe Bank and/or the HAT as possible structures hosting this possible subduction initiation. It would be much easier to explain (and in fact has already been proposed) than is suggested here.

**FIGURES**

**Figure 2A**

- the legend does not fit the map - green and blue lines.

- P-2 is missing (or I don't see it); P-3 is indicated 2 times; P-1 is wrongly indicated according to Fig 3 and the text…

-. I highly recommend adding more subdivisions between coordinates.

- I suggest to indicate in figure 2 the two segments of profile 5 shown in figure 3.

**Figure 3**

-Figure caption: It is not clear to which profile they refer in each case. This occurs for two reasons: 1) wrong nomenclature in Figure 2 (mentioned above); 2) mixes two nomenclatures "profile" and "a, b, c….", the latter not used in the figure 3.

- Figure caption: "Profile 1) Depth profile of seismicity along alignment North (Galicia) b) Profile along alignment south c)". According to Figure 2, these profiles are located to the south of both alignments.

- A complete legend is missing. For example, it is not indicated what the dark brown corresponds to, the two blues of the oceanic crust, the small red and purple dots.

-I also recommend indicating the extension of each segment (hyperextended, exhumed mantle...) in each profile since much reference is made to it in the text.

-Profile 4: there are 2 earthquakes in the water.

**L268 & Figure 4a:**

- If the Moho is the crust-mantle boundary and there is the ZECM (zone of Exhumed Continental Mantle) along the WIM, how can Figure 4 show the depth of the Moho in this zone if there is no Moho?

**Figure 5c:**

- By what name are the N-S thrusts represented in the central part of the WIM and in the SWIM known? and the long marine strike-slip fault at the latitude of Lisbon?

Sincerely,

Sara Martínez Loriente

**References:**

Baptista, M. A., Miranda, J. M., Batlló, J., Lisboa, F., Luis, J., and Maciá, R.: New study on the 1941 Gloria Fault earthquake and tsunami, *Nat. Hazards Earth Syst. Sci.*, 16, 1967–1977, https://doi.org/10.5194/nhess-16-1967-2016, 2016

Bartolome, et al.; Evidence for active strike-slip faulting along the Eurasia-Africa convergence zone: Implications for seismic hazard in the southwest Iberian margin. *Geology* 2012; 40 (6): 495–498. doi: https://doi.org/10.1130/G33107.1

Duarte, et al.; Are subduction zones invading the Atlantic? Evidence from the southwest Iberia margin. *Geology* 2013; 41: 839–842. doi: https://doi.org/10.1130/G34100.1

Geissler W.H. Matias L. Stich D. Carillho F. Jokat W. Monna S. Ibenbrahim A. Mancilla F. Gutscher M.-A. Sallarès V. Zitellini N., 2010, Focal mechanisms for sub-crustal earthquakes in the Gulf of Cadiz from dense OBS deployment: *Geophysical Research Letters* , v. 37, L18309, doi:10.1029/2010GL044289.

Gutscher, MA., et al, 2012. The Gibraltar subduction: A decade of new geophysical data, Tectonophysics, 574, 72-91; doi: 10.1016/j.tecto.2012.08.038

Martinez-Loriente, S., Sallares, V., and Gracia, E.:The Horseshoe abyssal plain thrust could be the source of the 1755 Lisbon earthquake and tsunami. Commun. Earth. Environ., 2, 145, doi:10.1038/s43247-021-00216-5, 2021

Martinez-Loriente, S. et al. Seismic and gravity constraints on the nature of the basement in the Africa-Eurasia plate boundary: new insights for the geodynamic evolution of the SW Iberian margin. J. Geophys. Res. Solid Earth 119, 127–149 (2014).

Ramos, A., Fernández, O., Terrinha, P. and Muñoz, J.A.: Extension and inversion structures in the Tethys-Atlantic linkage zone, Algarve Basin, Portugal, Int. J. Earth Sci., 105, 1663-1679, doi:10.1007/s00531-015-1280-1, 2016

Sallarès, V., et al.: Seismic evidence of exhumed mantle rock basement at the Gorringe Bank and the adjacent Horseshoe and Tagus abyssal plains (SW Iberia), Earth. Planet. Sci., 365, 120-131, doi:10.1016/j.epsl.2013.01.021, 2013.

---

## Referee Comment (RC2)

[referee-annotated manuscript omitted]

---

## Author Response (AR2)

Answer to RV1:

| COMMENTS BY MARTINEZ-LORIENTE | ANSWER, CHANGES IN MANUSCRIPT BY FERNANDEZ-VIEJO ET AL |
|---|---|
| L45-47: If the "W-E thrusts" refers to the Gorringe Bank, change it to "NE-SW thrusts" | We agree. Corrected. |
| L59: The Africa-Europe plate boundary continues east of the Strait of Gibraltar. I suggest to rephrase the expression | We have revised the sentence to highlight the diverse characteristics along the entire plate boundary. It is important to note that the boundary extends east of Gibraltar, signifying its complex nature throughout. |
| L62: Add space "theWIM" | Ok. Corrected. |
| L62: what is the "WIM abyssal plain"? If it exists, locate it in figure 1. | It refers to the oceanic seafloor of the margin, where anomalies are used for kinematic reconstructions. We have rephrased and put the Iberian Abyssal plain as the name to refer to this extended area |
| L63: Add "." after ")" | Ok. Corrected. |
| L70: Ramos et al., 2016 focused on the Algarve basin, southern Portugal… nothing related to structural domains along the WIM. | We agree, it was a mistake. Corrected. |
| L78-80: Sallarès et al., 2013 and Martínez-loriente et al., 2014 presented geophysical evidence suggesting that the Gorringe Bank and the neighbouring abyssal plains are composed mainly of exhumed mantle rocks and presented a new model for the opening of the North Atlantic. I suggest including these references. | We have avoided to explain in the introduction the southernmost area of the Iberian margin, due to the fact that its seismicity cannot be considered intraplate, but related to the plate boundary. Therefore, we do not include most of the works from this area. In any case, those references have been used later when appropriate in the discussion |
| L87-114: I highly recommend adding a map with the domains referenced in this section, and the delimitation of the different segments of the WIM. It is very difficult to follow the (messy) description of the authors. Therefore, I also suggest rewriting the section | Yes, we have rewritten the section that was exposed in an unclear way. We have also corrected the corresponding figures and references. |
| L90-92 & Fig 2: "The OCT extends between 12º10'W and 12º30'W in the IAP (Whitsmarsh et al., 1990) and it would extend Nº10 for 130 km until Extremadura Spur (…)" It is very difficult to correctly locate the COT with the map coordinates. I highly recommend adding more subdivisions between coordinates. How can readers locate longitude 12º10'W if all the information they have is the location of longitudes 12ºW and 16ºW? | We agree, we have modified the coordinate axes of the figures, which now have intervals every 0.5 degrees. |
| L91: 10ºN, Sure? | It is indeed a mistake. It should read N70ºE. |
| L94: "The exhumed mantle domain has been drilled at a serpentinite ridge (Boillot et al., | The location of the OPD drilling is now included in Figure 2 and figure description. |

| | |
|---|---|
| 1988b)" Where? | |
| L95-98. Add reference. | Done, it is from Amigo-Marx, 2022 |
| L100: DGM? It is not located on the map and the abbreviation is not described | We have added the location and name of this area (from Granado et al., 2021) |
| L102: "The GB is 15-20 km thick"… and? Made of?... | Certainly. Apologies for the inadvertent deletion in the previous sentence. The revised version has been corrected to address the issue. |
| L107: What is "THD" y "dZ"? where are these maps? | THD = Total horizontal Derivative; dZ = Vertical derivative; In both cases referring to maps obtained by Granado et al, 2021. This has been better phrased in the new version as it was misleading and those maps are not shown in this manuscript but only referred to. Corrected |
| L111-114: Let's see, if the WIM is divided into 3 segments, and the authors say that there is a segment further south than what they call "South WIM", wouldn't it be more logical to call it Central WIM or something similar? | Yes, it makes more sense to rename the segments in this way. We have done it now in the manuscript, taking the whole margin as divided in three distinct segments. This study only deals with the North and Central segments as the seismicity in the third southernmost segment is too close to the plate boundary to be considered intraplate |
| L116-120: when? it should be specified to which period the authors refer | A sentence has been added to specify the period where Iberia functioned as an independent plate according to some authors. |
| L135-136: when? At present? At the beginning? Has it been constant over time? | Well, the authors refer those numbers "since the Miocene", which implies up to now but they do not say if it has been constant or not over time, which is also quite difficult to prove with that methodology only. |
| L138-142: I suggest the authors include some arrows in figure 1 that indicate the kinematics along the plate boundary. | Done |
| L142: Again, Ramos et al. only investigate a small portion of the SW Iberian Margin, the Algarve basin south of Portugal. There are many other works that propose the reactivation of thrusts throughout SW Iberia at a regional level, such as Martinez-Loriente et al. 2013 (but there are many others). Ramos's work is very local and their conclusions quite debatable. | We have added a cite for this sentence. |
| L144: The Gloria Fault is the source of one of the largest earthquakes occurred in the North Atlantic, 1941 Mw 8.3-8.4 (e.g., Baptista et al., 2016) | Yes, this paragraph was incomplete. We have made the necessary revisions by rephrasing it and incorporating additional references to ensure its accuracy and comprehensiveness. |
| L146-152: In this section the authors mix nanostrain/yr and mm/yr... for non-experts, it is difficult to compare the different geodetic velocities. | We have added the nomenclature of nanostrain in mm/yr in parenthesis for clarity (1 mm/yr/1000 km → 1 nanostrain/yr) By definition, strain is a relative change in distance, divided by the distance over which the change occurs, for example, 1 mm change in 1 km long line corresponds to a strain change of 1 part in $10^6$ or 1 microstrain. It relates to a different concept than a simple displacement of 1mm/yr, which means a velocity |

| | (distance over time). Therefore, it is not mixing, just giving the units that were used in the references and their significance. |
|---|---|
| L179: Why have you included historical seismicity but not the instrumental seismicity available prior to 2003? What's the point of including the first and ignoring the second? I would like to know what these 10 historical earthquakes are, and if there is any relevant aspect, that it be recorded in the figures, text, in a table (somehow). | The fact of taking data from 2003 is not arbitrary. Between 2001 and 2003 the IGN network changed and data became more reliable, according to an increased number of stations and also to the fact that stations passed to have three components instead just one. Chasing a higher quality of the data was the main purpose of this decision. On historical earthquakes, they really do not contribute to the results of this study apart from the fact that they do exist in the area. They could be introduced in the figures if reviewer think is a significant aid, but we honestly have the idea that they will only increase the density of the images and not contribute to clarity or evidences. They are the green dots portrayed below

 |
| L182: "The 9 focal mechanisms considered in this work have been obtained from the CMT Catalogue". Figure 2 includes 14 focal mechanisms (not 9 as mentioned by the authors) | Yes, we gathered 14, but only 9 of them are in the marine area. We have rephrased that to make it clear.… |
| L185: "(ii) geological structures mapped in the continental platform (Somoza et al., 2021)". Why only those included in the continental shelf? | The sentence was misleading, because we have included other structures interpreted in the non-continental area of the margin. We rephrased. |
| L195: "Further up towards the MAR, there is some isolated events.". Could this lack of seismicity be associated with the distance to the onshore stations? | This issue does not stem from a problem with the level of detection or sensitivity. Instead, it is characterized by a gradual decrease in detection as distance increases. Notably, events are detected towards the south, even when they occur at the same or greater distance from the monitoring stations. Additionally, the stations successfully capture seismic activity from the abundant mid-oceanic ridge, which is further away from the study area. Hence, based on these observations, the answer is no |
| L196: "The orientation of these bands is about N80ºW". In my opinion, the southern alignment has a clear E-W orientation and it is related to the Estremadura Spur and the Tore Seamount (it is not located on figure 1). | The determination of the overall orientation relies on the solutions derived from the density map. While it is indeed noted in the subsequent text that events in the proximal margin adhere to the Spur, the width of this cluster raises doubts regarding the direction of the band's lineation. In any case, this deviation is only 10 º from the E-W direction that the reviewer perceives. We have now included the Tore seamount in Figure 1. |

| | |
|---|---|
| L202: "This density band of events is oriented N75ºW from the coast until 17.5ºW". I disagree with the interpretation of this alignment. There is an E-W alignment from the coast until 12ºN related to the Galicia Bank and a second cluster of seismicity to the northwest related to another ridge/ relief (without name in Figure 1). Between both clusters of seismicity there is a gap of more than 100 km without seismicity, so there is no evidence to indicate a relationship between them. | Yes, we do acknowledge the paragraph in the discussion section that highlights this fact. However, as mentioned earlier, when constructing the density map, the overall trend of both separate clusters is considered, which guides the determination of these directions based on specific criteria. While it is possible to subdivide the clusters into slightly different trends, the global perspective provided by the density map solution aims to fit all the events into the most reliable and cohesive solution as a whole. The gap of seismicity in the northern lineation is evident, while still the density map shows both clusters follow the direction explained in the text Additionally, we have taken your suggestion into account and have included the Coruña Seamount in Figure 1. |
| L211: "The number of events is larger at the transition between hyperextended crust and exhumed mantle" As it is not indicated in figure 3, I do not know what the authors consider to be the hyperextended crust and the exhumed mantle domains. According to my consideration (which coincides with that of Granados et al., where the profiles come from), there is exactly 1 earthquake in this segment (Profile 1). Therefore, I think the authors' statement is wrong. | The location of that unique earthquake is in the transition between exhumed mantle and oceanic domains. We corrected the statement. The majority of the events in this cluster are situated below the necking and the hyperthinned domains (brown in the figure) towards the exhumed mantle (green) domain. According to the domain map we can say that the number of events is larger at the transition between the necking and the hyperthinned domains. |
| L211: "There is an arguably but noticeable 50 km wide gap in event distribution west of the Galicia Bank". Why this gap is "arguably"? In 150 km there are exactly 2 earthquakes. | Yes, we have rephrased according to your feedback. It is evident. |
| L2014-2015: "especially within the transition between the hyperextended and exhumed mantle domains". Same as in the comment of Line 211. | Refer to the answer of previous comment. Figures have been modified to include the limits of passive margin domains and a legend with colours to differentiate them. |
| L215-216: "A particular set of south-dipping earthquakes can be observed in profile 5". This is highly debatable. The seismicity could be vertically aligned, or even dip to the southwest but with a lower dip than that interpreted in figure 3 by the authors. | We appreciate your observation. Upon reviewing profile 5a, specifically the first panel below the Galicia margin, it appears evident that the events deepen towards the south, forming an inclined wedge. The base of this wedge has a lesser dip than initially indicated, as you pointed out correctly. In contrast, the second panel below the Extremadura spur does indeed exhibit vertical alignments. We acknowledge that this distinction may not have been clear initially, and therefore, we have taken steps to address this concern. We have rephrased the relevant text and included profile 5a and profile 5b in both the figures and the accompanying text to provide further clarification. |
| L218: "Some of the focal mechanism in this area…". There is a lot of distance between the few focal mechanisms shown in figure 2 and this seismicity. By the way, why are some focal mechanisms represented in red and others in blue? It is not indicated in the legend or in the figure caption | The red ones correspond to data from IGN, the blue ones to the ones obtained through CTM. Added the clarification in the figure |
| L219-221: As I mentioned before, I see this seismicity aligned E-W from the coast up to 14 or 15ºN (? = it is difficult for me to be precise with the low coordinate discretization of | The partitions of the axis of the figures have been incremented for better understanding. Again, N80W is 10 degrees shorter than E-W direction; we have chosen that number based on the density map solutions. |

| | |
|---|---|
| figure 2) | |
| 228-230. In this case, I agree that there is an amount of seismicity in the transition between hyperextended crust and exhumed mantle, but I disagree with "a lack of them in the oceanic crust until the western termination…" the seismicity decreases, but there are 15 or 20 earthquakes. | We have rephrased, as "lack" implies too absolute and it is not true. Changed |
| L231: It would be interesting to know which segment of profile 5 of Figure 2 is represented in Figure 3, since it would allow to locate on the map these possible vertical seismicity alignments. | Yes, we have added the locations of both profile 5a and profile 5 b within the long line in a wider legend |
| L234-235: "… the highest magnitude earthquakes in this area occur in the subcrustal mantle and below two seamounts…" What??? In profile 4 the seismicity is projected (100 km). If we look at the map (figure 2), these 2 earthquakes that the authors refer to are located far from these two seamounts or volcanic edifices. | Yes, that is right. Maybe the projection of event gives a misleading picture. However, the location of the events and their magnitude suggest some relation to the volcanic nature of the topographic highs, |
| 235-236: "The referred vertical alignment would be consistent with a volcanic origin for those particular events." Are the authors referring to the vertical alignment mentioned in the previous paragraph (231-233)? If so, it is difficult for me to understand the relationship that the authors see between this seismicity of the Estremadura spurn that is seen in the southern part of profile 5 with these two earthquakes that are seen in profile 4 and that the authors say are related to two volcanic edifices (which actually aren't)? | We have reprased |
| L254-256: Geissler et al. 2010 already showed that in SW Iberia the majority of seismicity occurred between 40-60 km depth, and with strike-slip or inverse focal mechanism solutions. Bartolomé et al. (2012) associated the strike-slip seismicity with the Lineament North and Lineament South strike-slip faults. Martínez-Loriente et al. (2021) associated the deep inverse seismicity as well as the largest seismic events occurred in the region with the HAT. | We include the reference with the hypocentral depths of those events in the area adding to the text of this paragraph. |
| L256: "?" Delete it | Done |
| 279: "Seismicity almost abruptly stops around the area of undisputed oceanic crust. Nonetheless, there is still a few events westward toward the MAR, and they follow the N80ºW direction too.". In the north, the seismicity stops just before the COB (around 12ºW), more than 50 km before the oceanic crust. In the south, seismicity does not stop at any point and continues from one domain to another | In figure 4d according to the domain map seismicity occurs in the oceanic crust north and south. Undisputed oceanic crust is marked by anomaly M3 which according to most authors is a real oceanic magnetic anomaly, unlike anomaly J which is disputed. In any case, we have rephrased slightly the paragraph as in the vertical profiles, looks like a more abrupt change than in the map. |

| | |
|---|---|
| L290: in figure 2 it does not include the AGFZ, so I cannot get a visual idea of what the distance is between it and the southern alignment. | Right. We have make the reference to Figure 1 which includes the AGZF now. |
| L310: "GAP"? Describe the abbreviation | GAP in seismology is the maximum angle separating two adjacent seismic stations, both measured from the epicenter of an earthquake |
| L343-346: There are a lot of scientific publications showing strike-slip faults with MCS data. I can include 10 or 15 references only in the SW of Iberia. I know that the authors have access to seismic profiles acquired in the WIM. If they don't see the strike-slip faults, could it be that these structures don't really exist? | With that paragraph we only want to point out that a pure strike slip fault as with any vertical structure would be impossible to discern in near vertical incidence seismic profiles unless the blocks in contact show different types of basement or geological history, and even less in oceanic domains with only one type of rock. If they have a mixed component of transpression or transtension they may be identified in seismic lines, but again, if the sediments on top show any dislocations at the resolution level of the seismic wave. We are not aware or have seen in any of the profile such a structure in this margin. Therefore, although we do not have evidences, we cannot conclude that they do not exist. |
| | It is logical that the may be seen in SW Iberia close to the plate boundary where actual deformation is taking place quickly and constantly. Within the central and northern parts of the margin, it may be quite difficult to identify any structure if its activity is low or spaced in time. We are still in a passive quiet margin. Seismicity would be the first indication of activity and incipient formation of these structures. Therefore, what we meant in the text is that seismicity implies some strain and deformation is going on, so maybe some incipient strike slip structures are being formed, as focal mechanisms suggest that the release of stress is taking place along that type of discontinuities. |
| L408: "NE-SW thrust systems extending 300 km along the WIM accommodate the arc-orthogonal convergence (Gutscher et al., 2012)". What are these fault systems??? Specify them and add references. Gutscher et al., investigated the possible subduction under the Gibraltar Arc, nothing related to the WIM or any "thrust system" there. | We have rewritten and clarified this paragraphs |
| L409: "and younger thrust faults are nucleating along the west Portuguese passive margin or in the Tagus Abyssal plain". Which ones? Specify them and add references where the existence of these structures can be verified. The work of Duarte et al. (2013) does not count as a reference since they only presented a theory without a single real data to support it. | We have added the pertinent references |
| L416-428: I am surprised that the authors do not consider the Gorringe Bank and/or the HAT as possible structures hosting this possible subduction initiation. It would be much easier to explain (and in fact has already | We have added those references, but we do not deal with the southernmost part of the WIM. We wanted to focus on the NOT plate boundary   seismicity and how these events can relate to the different theories in the literature. We do not postulate or support a subduction start here or there, we just |

| | |
|---|---|
| been proposed) than is suggested here. | want to express that the WIM seismicity within the central and northern parts of the margin does not support a subduction interface forming along the WIM north of the actual plate boundary. |
| **Figure 2A**

- the legend does not fit the map - green and blue lines.

- P-2 is missing (or I don't see it); P-3 is indicated 2 times; P-1 is wrongly indicated according to Fig 3 and the text…

-. I highly recommend adding more subdivisions between coordinates.

- I suggest to indicate in figure 2 the two segments of profile 5 shown in figure 3. | The mistakes have been corrected and subdivisions in geographical coordinates increased; also indicated the profiles 5a and 5b now |
| **Figure 3**

-Figure caption: It is not clear to which profile they refer in each case. This occurs for two reasons:

1) wrong nomenclature in Figure 2 (mentioned above); 2) mixes two nomenclatures "profile" and "a, b, c….", the latter not used in the figure 3.

- Figure caption: "Profile 1) Depth profile of seismicity along alignment North (Galicia) b) Profile along alignment south c)". According to Figure 2, these profiles are located to the south of both alignments.

- A complete legend is missing. For example, it is not indicated what the dark brown corresponds to, the two blues of the oceanic crust, the small red and purple dots.

-I also recommend indicating the extension of each segment (hyperextended, exhumed mantle...) in each profile since much reference is made to it in the text.

-Profile 4: there are 2 earthquakes in the water. | Mistakes have been corrected and improved following the reviewers notes.
We have added the legend and extension of rift domains along the profiles

- |
| **L268 & Figure 4a:**

- If the Moho is the crust-mantle boundary and there is the ZECM (zone of Exhumed Continental Mantle) along the WIM, how can Figure 4 show the depth of the Moho in this zone if there is no Moho? | Well, the Moho is the discontinuity (at present time) that separates crust from lithospheric mantle. It does exist beneath all kind of domains and oceanic crust except beneath mid oceanic ridges where lithospheric mantle is being erupted at the surface. And it is defined by a change in seismic velocity, observed in refraction/wide angle reflection profiles, normally from values of 6.8-6.9 to 8.0-8.3 in continental crust, and values from 7.5-7.8 to over 8.0 in zones where peridotites or hydrated, exhumed mantle, serpentinites, are present.

Although in the MCS data it is not observed a reflectivity that can be Moho, in all the refraction/WAR lines crossing the central and north Iberian margin, Moho has been interpreted based on PmP reflections, In some cases beneath the interpreted exhumed mantle Vp shows a high gradient to reach the velocities of mantle and Moho is not a first order discontinuity. (Afilhado et al 2008, Dean et al, 2000) If this is what the reviewer is referring too, this fact does not mean that Moho does not exist. Lithospheric mantle is interpreted |

| | generally when Vp reach 8.0km/s or higher. So, the map of crustal thickness reflect this velocity change, independently that the Moho appears as a first order discontinuity or as a change in velocity gradient

. |
|---|---|
| **Figure 5c:**
- By what name are the N-S thrusts represented in the central part of the WIM and in the SWIM known? and the long marine strike-slip fault at the latitude of Lisbon? | The image has been taken from a publication where those names are not present, it was just to illustrate the possible way to start a subduction along structures that may or may not be present. |

**Response to Reviewer 2**

We appreciate the time and effort invested on reviewing the work and thank you for the suggestions, comments and the editing. We have now made the pertinent changes included in the revised version. Most of them were accepted, except from paragraphs were other changes were included following the suggestions and comments of a previous revision.

The question aside from the text that the reviewer posts is if there would be a "radial" rather than linear pattern in the seismicity that seems to coincide with the pattern of a magnetic anomaly in the area (fig 4c).

The answer is: possibly. Although at large distance the striped NW-SE pattern seems quite clear, it is also acknowledgeable that in closer inspection the striped trend gets diffuse, adding difficulty on the interpretation and grouping events in smaller clusters. The magnetic anomaly that the reviewer refers to, results of the structural disposition of the basement rocks, the variscan formations that can be seen on land. Our understanding is that probably the pattern of seismicity presents more complication than the simplistic NW-SE bands we talk about, something that it is intrinsic to the study of this type of moderate, low seismicity in intraplate "quiet" settings. Therefore, any minor structure slightly moving, added to the uncertainty location, will give events that may be clustered in specific smaller scale structural features such as the one the magnetic anomaly evidences.

Regarding the text edits and comments in the manuscript, we proceed to deal with all of them individually.

Line 152 comment on "the map shows in this area". It is an unfortunate phrasing. We have changed the text to serve the meaning we wanted to give to this sentence.

Line 198 "however the separation in two bands is not that simple". Well, this is a poorly expressed explanation. Please refer to the introduction of this document. We have rephrased.

Line 218 "inverse fault" changed to reverse fault.

Line 226 "which relates to an unknown error". This refers to the fact that the depth location of seismicity has always a higher error than in the XY coordinates. This is due to the method for getting the events depth, which uses as input a preliminary velocity model. In turn, this velocity model and its variations affect very much the depth of solutions and the more geologically restricted the better. In this area, where there is an abundant set of deep seismic reflection and refraction data, the local velocity model is as good as it can be, but it also has its own depth-velocity errors.

Line 232 "wihin it". To the north of it.

Line 235 "the referred vertical alignment would be consistent with a volcanic origin for those particular events". The reviewer says (rightly) that there is not vertical alignment in this location. The reviewer number one also pointed out this contradiction. We have amended those paragraphs, but still noting that some vertical alignments may have to do with volcanism as the area shows an abundance of these edifices in close areas, preferring this interpretation to a structural cause, due to the mantle depth of most of the events.

Line 275 "main N-S disposition of the magnetic anomalies". The reviewer points out the circular pattern in the magnetic anomaly, which seems to cluster also the events. And yes, the reviewer is right about that, we have partially answered or commented this fact at the beginning of the document and also in the text. The phrase refers more to the N-S alignment of the oceanic magnetic anomalies, which are parallel to the ridge. Of course, in the continental platform and margin, the complexity of the magnetic anomalies has to do with the continental basement. The seismicity seems to cluster around several structural features within the margin but we did not find a clear relation to known structures or outcropping features in the sea floor.

Line 299 "the text highlights the limitations of seismotectonic interpretations due to inconclusive mapping of structures in the sea floor of the WIM". We have rewritten this paragraph and tried not to repeat the things, moving them to section 2.

Line 227 "may have been bouncing from its perpendicular to extension direction". This means that the Iberian Peninsula, specifically its N-S Margin along Spain and Portugal, which now is parallel to the ridge, may have been oblique to the ridge extension direction in several periods while the Atlantic opening took place. This means the micro Iberian plate included some type of rotation adding complexity to the margin structure. The presence of the triple point to the north also supports the evidences of rotational movement of Iberia while extension along the ridge maintained its direction.

The revised manuscript with the changes is uploaded in the system for its reassessment.

Thank you, sincerely,

The Authors

[revised manuscript text omitted]

